# Learning Sampling Distributions for Model Predictive Control

**Jacob Sacks**
University of Washington
jsacks6@cs.washington.edu

**Byron Boots**
University of Washington
bboots@cs.washington.edu

**Abstract:** Sampling-based methods have become a cornerstone of contemporary approaches to Model Predictive Control (MPC), as they make no restrictions on the differentiability of the dynamics or cost function and are straightforward to parallelize. However, their efficacy is highly dependent on the quality of the *sampling distribution itself*, which is often assumed to be simple, like a Gaussian. This restriction can result in samples which are far from optimal, leading to poor performance. Recent work has explored improving the performance of MPC by sampling in a learned latent space of controls. However, these methods ultimately perform all MPC parameter updates and warm-starting between time steps in the control space. This requires us to rely on a number of heuristics for generating samples and updating the distribution and may lead to sub-optimal performance. Instead, we propose to carry out all operations in the latent space, allowing us to take full advantage of the learned distribution. Specifically, we frame the learning problem as bi-level optimization and show how to train the controller with backpropagation-through-time. By using a normalizing flow parameterization of the distribution, we can leverage its tractable density to avoid requiring differentiability of the dynamics and cost function. Finally, we evaluate the proposed approach on simulated robotics tasks and demonstrate its ability to surpass the performance of prior methods and scale better with a reduced number of samples.

**Keywords:** Model Predictive Control, Normalizing Flows

## 1 Introduction

Sequential decision making under uncertainty is a fundamental problem in machine learning and robotics. Recently, model predictive control (MPC) has emerged as a powerful paradigm to tackle such problems on real-world robotic systems. In particular, MPC has been successfully applied to helicopter aerobatics [1], robot manipulation [2, 3, 4], humanoid robot locomotion [5], robot-assisted dressing [6], and aggressive off-road driving [7, 8, 9]. Sampling-based approaches to MPC are becoming particularly popular due to their simplicity and ability to handle non-differentiable dynamics and cost functions. These methods work by sampling controls from a policy distribution and rolling out an approximate system model using the sampled control sequences. They then use the resulting trajectories to compute an approximate gradient of the cost function and update the policy. Next, the controller samples an action from this distribution and applies it to the system. It repeats the process from the resulting next state, creating a feedback controller. Between time steps, it warm-starts the optimization process using a modification of the solution at the previous time step.

An important design decision is the form of the sampling distribution, which is often simple, e.g. a Gaussian, such that we can efficiently sample and tractably update its parameters. However, this also has drawbacks: without much control over the distribution form, samples often lie in high-cost regions, hindering performance. This can be particularly problematic in complex environments with sparse costs or rewards, as a poorly parameterized distribution may hinder efficient exploration, leading the system into bad local optima. A side effect is that we often require many samples to accomplish the objective, increasing computational requirements. There have been extensions which target more complex distributions, such as Gaussian mixture models [10] and a particle method based on Stein variational gradient descent (SVGD) [11]. However, there is a large amount of structure in the environment that these methods fail to exploit. Instead, an alternative approach is to *learn* a sampling distribution which can leverage the environmental structure to draw more optimal samples.

6th Conference on Robot Learning (CoRL 2022), Auckland, New Zealand.

Prior work on learning MPC sampling distributions generally requires differentiability of the dynamics and cost function [12, 13]. Power and Berenson [14, 15] circumvent this issue by leveraging normalizing flows (NFs) [16, 17, 18], which have a tractable log-likelihood. This property allows them to learn flexible distributions by directly optimizing the MPC cost without requiring differentiability via the likelihood-ratio gradient. However, a limitation of their approach is that all online updates to the distribution and warm-starting between time steps occur entirely in the control space, leaving the latent distribution fixed. This forces them to apply heuristics to generate samples by combining those from the learned distribution with Gaussian perturbations to the current control-space mean. These restrictions prevent us from fully taking advantage of the learned distribution and potentially throws away useful information. Additionally, their approach does not allow for the incorporation of control constraints directly into the sampling distribution, which is important for real-world robots.

Instead, we propose to alter the optimization machinery to operate entirely in the latent space. As the NF latent space follows a simple distribution, it remains feasible to perform MPC updates in this learned space and update the latent distribution online. Specifically, during an episode, the parameters of the latent distribution are updated with MPC while those of the NF remain fixed. Then during training, after each episode, the parameters of the NF are updated. We can frame this setup as a bi-level optimization problem [19] and derive a method for computing an approximate gradient through the latent MPC update. This involves treating MPC as a recurrent network, where the control distribution acts as a form of memory, and unrolling the computation to train with backpropagation-through-time (BPTT). However, it is no longer clear how to warm-start between time steps because there is no clear delineation of time in the latent space. Moreover, the usual method of warm-starting, which simply shifts the current plan forward in time, may be sub-optimal. Therefore, we propose to learn a shift model, which performs all warm-starting operations in the latent space. Finally, we show how to alter the NF architecture to incorporate box constraints on the sampled controls.

**Contributions:** In this work, we build upon recent efforts to learn sampling distributions for MPC with NFs by moving all online parameter updates and warm-start operations into the latent space. We accomplish this by framing the learning problem as bi-level optimization and derive an approximate gradient through the MPC update of the latent distribution to train the network with BPTT. Additionally, we show how to parameterize the flow architecture such that we can incorporate box constraints on the controls. Finally, we empirically evaluate our proposed approach on simulated navigation and manipulation tasks. We demonstrate its ability to improve performance over the baselines by taking full advantage of the learned latent space. We find that the performance of the controllers with our learned sampling distributions scales more gracefully with a reduction in the number of samples.

## 2  Sampling-Based Model Predictive Control

We consider the problem of controlling a discrete-time stochastic dynamical system, which is in state $x_t \in \mathbb{R}^N$ at time step $t$. Upon the application of control $u_t \in \mathbb{R}^M$, the system incurs the instantaneous cost $c(x_t, u_t)$ and transitions to $x_{t+1} \sim f(x_t, u_t)$. We wish to design a policy $u_t \sim \pi(\cdot|x_t)$ such that the system achieves good performance over $T$ steps. Instead of finding a single, globally optimal policy, MPC re-optimizes a local policy at each time step by predicting the system's behavior over a finite horizon $H < T$ using an approximate model $\hat{f}$. Specifically, in sampling-based MPC, we sample a control sequence $\hat{\boldsymbol{u}}_t \sim \pi_\theta(\cdot)$, where $\hat{\boldsymbol{u}}_t \triangleq (\hat{u}_t, \hat{u}_{t+1}, \cdots, \hat{u}_{t+H-1})$, and our policy is parameterized by $\theta \in \Theta$. We rollout our model starting at $x_t$ using these sampled controls to get our predicted state sequence $\hat{\boldsymbol{x}}_t \triangleq (\hat{x}_t, \hat{x}_{t+1}, \cdots, \hat{x}_{t+H})$, with $\hat{x}_t = x_t$. The total trajectory cost is

$$C(\hat{\boldsymbol{x}}_t, \hat{\boldsymbol{u}}_t) = \sum_{h=0}^{H-1} c(\hat{x}_{t+h}, \hat{u}_{t+h}) + c_{term}(\hat{x}_{t+H}), \tag{1}$$

where $c_{term}(\cdot)$ is a terminal cost function. We then construct a statistic $\hat{J}(\theta; x_t)$ defined on cost $C(\hat{\boldsymbol{x}}_t, \hat{\boldsymbol{u}}_t)$ and use the rollouts to solve $\theta_t \leftarrow \arg\min_{\theta \in \Theta} \hat{J}(\theta; x_t)$. After optimizing our parameters, we sample the control sequence $\hat{\boldsymbol{u}}_t \sim \pi_{\theta_t}(\cdot)$, apply the first control to the real system (i.e. $u_t = \hat{u}_t$), and repeat the process. Because each parameter $\theta_t$ depends on the current state, MPC effectively yields a state-feedback policy, even though the individual distributions give us an open-loop sequence.

In this paper, we consider a popular sampling-based MPC algorithm known as Model Predictive Path Integral (MPPI) control [7, 8]. MPPI optimizes the exponential utility or risk-seeking objective:

$$\hat{J}(\theta; x_t) = -\log \mathbb{E}_{\pi_\theta, \hat{f}} \left[ \exp\left( -\frac{1}{\beta} C(\hat{\boldsymbol{x}}_t, \hat{\boldsymbol{u}}_t) \right) \right], \tag{2}$$

where $\beta > 0$ is a scaling parameter, known as the temperature. As we do not assume that the dynamics or cost function are differentiable, we compute the gradients via the likelihood-ratio derivative:

$$\nabla \hat{J}(\theta; x_t) = -\frac{\mathbb{E}_{\pi_\theta, \hat{f}}\left[e^{-\frac{1}{\beta}C(\hat{\boldsymbol{x}}_t, \hat{\boldsymbol{u}}_t)}\nabla_\theta \log \pi_\theta(\hat{\boldsymbol{u}}_t)\right]}{\mathbb{E}_{\pi_\theta, \hat{f}}\left[e^{-\frac{1}{\beta}C(\hat{\boldsymbol{x}}_t, \hat{\boldsymbol{u}}_t)}\right]}. \tag{3}$$

In MPPI, the policy is assumed to be a factorized Gaussian of the form

$$\pi_\theta(\hat{\boldsymbol{u}}) = \prod_{h=0}^{H-1} \pi_{\theta_h}(\hat{u}_{t+h}) = \prod_{h=0}^{H-1} \mathcal{N}(\hat{u}_{t+h}; \mu_{t+h}, \Sigma_{t+h}). \tag{4}$$

Previous work by Wagener et al. [9] has shown that optimizing this objective with dynamic mirror descent (DMD) [20] and approximating with Monte Carlo estimates gives us the MPPI update rule:

$$\mu_{t+h} = (1 - \gamma_t)\tilde{\mu}_{t+h} + \gamma_t \sum_{i=1}^{N} w_i \hat{u}_{t+h}^{(i)}, \quad w_i = \frac{e^{-\frac{1}{\beta}C(\hat{\boldsymbol{x}}_t^{(i)}, \hat{\boldsymbol{u}}_t^{(i)})}}{\sum_{j=1}^{N} e^{-\frac{1}{\beta}C(\hat{\boldsymbol{x}}_t^{(j)}, \hat{\boldsymbol{u}}_t^{(j)})}} \tag{5}$$

where $\tilde{\mu}_{t+h}$ is the current mean for each time step and $\gamma_t$ is the step size. Between time steps of DMD, we get $\tilde{\mu}_{t+h}$ from our previous solution $\mu_{t+h}$ by using a shift model $\tilde{\mu}_{t+h} = \Phi(\mu_{t+h})$. This shift model aims to predict the optimal decision at the next time step given the previous solution. In the context of MPC, it allows us to warm-start the optimization problem to speed up convergence, as we can only approximately solve the optimization problem due to real-time constraints.

## 3 Learning the Sampling Distribution of MPC

### 3.1 Representation of the Learned Distribution

Instead of using uninformed sampling distributions, learned distributions can potentially exploit structure in the environment to draw samples which are more likely to be collision-free and close to optimal. However, such learned distributions must be sufficiently expressive in order to better capture near-optimal, potentially multimodal, behavior. They must also be parameterized such that it is tractable to sample from and update online. If the distribution has a large number of parameters, the number of samples required to efficiently update them online may be computationally infeasible. And ideally, the form of our distribution would be such that we could find a closed-form update.

One path towards meeting these criteria is to maintain a simple latent distribution from which we can sample, and then learn a transformation of the samples which maps them to a more complex distribution. During training, we learn the parameters of this transformation, which can be conditioned on problem-specific information, such as the starting and goal configurations of the robot and obstacle placements. However, when executing the policy during an episode, the parameters of this learned transformation remain fixed, and instead, we update the parameters of the latent distribution. Concretely, we consider learning a distribution $\pi_{\theta, \lambda}$ which is defined implicitly as follows:

$$\hat{\boldsymbol{z}}_t \sim p_\theta(\cdot), \quad \hat{\boldsymbol{u}}_t = h_\lambda(\hat{\boldsymbol{z}}_t; c) \tag{6}$$

where $\hat{\boldsymbol{z}}_t \triangleq (\hat{z}_t, \hat{z}_{t+1}, \cdots, \hat{z}_{t+H-1})$, $c$ is a context variable describing the relevant information of the environment, $p_\theta$ is the latent distribution with parameters $\theta$, and $h_\lambda$ is the learned conditional transformation with parameters $\lambda$. Moving forward, we assume that both $\hat{\boldsymbol{z}}_t$ and $\hat{\boldsymbol{u}}_t$ are stacked as vectors in $\mathbb{R}^{MH}$. If $p_\theta$ is a Gaussian factorized as in Equation (4) and we assume that $h_\lambda$ is invertible, we prove in Appendix A.6 that the corresponding DMD update to the latent mean is simply Equation (5), except that we replace the controls in the weighted sum with the latent samples.

### 3.2 Formulating the Learning Problem

Learning the distribution $\pi_{\theta, \lambda}$ amounts to solving a bi-level optimization problem [19], in which one optimization problem is nested in another. The lower-level optimization problem involves updating the latent distribution parameters at each time step, $\theta_t$, by minimizing the expected cost with DMD. The upper-level optimization problem consists of learning $\lambda$ such that MPC performs well across a number of different environments. To formalize this, first consider that we have some distribution of

environments $c \sim \mathcal{C}(\cdot)$ over which we wish MPC to perform well. For each environment, our system has some conditional initial state distribution $x_0 \sim \rho(\cdot|c)$. The objective we wish to minimize is then

$$\ell(\boldsymbol{\theta}, \lambda; c) = \mathbb{E}_{\boldsymbol{\pi}_{\boldsymbol{\theta}, \lambda}, \rho, f} \left[ \sum_{t=0}^{T-1} \hat{J}(\theta_t, \lambda; x_t, c) \right] \tag{7}$$

where $\boldsymbol{\theta} = (\theta_0, \theta_1, \cdots, \theta_{T-1})$ and our cost statistic, $\hat{J}$, now depends on $\lambda$ and $c$ as well. This objective measures the expected performance of the intermediate plans produced by MPC along the $T$ steps of the episode. Our desired bi-level optimization problem can be formulated as:

$$\min_{\lambda} \ \mathbb{E}_{\mathcal{C}} \left[ \ell(\boldsymbol{\theta}(\lambda), \lambda; c) \right] \quad \text{s.t.} \ \boldsymbol{\theta}(\lambda) \approx_{\lambda} \arg\min_{\boldsymbol{\theta}} \ell(\boldsymbol{\theta}, \lambda; c) \tag{8}$$

where $\approx_{\lambda}$ indicates that we approximate the solution of the optimization problem with an iterative algorithm that may also be parameterized by $\lambda$, as the exact minimizer is not available in closed form. Moreover, the notation $\boldsymbol{\theta}(\lambda)$ indicates the dependence of the lower-level solution on the upper-level parameters. In our case, we solve the lower-level problem with DMD, where we also parameterize the shift model, $\Phi_{\lambda}(\cdot; c)$, making it a learnable component and conditioned on $c$.

The normal shift model in MPC simply shifts the control sequence forward one time step and appends a zero or random control at the end. However, because we are performing this update in the latent space, there is no clear delineation between time steps of the latent controls, as they are coupled according to the learned transformation. Therefore, there is no way to easily perform the equivalent shift operation in the latent space. As such, we instead learn this shift model along with the transformation. Besides, the standard approach described above may not be optimal. By learning it, we may be able to further improve performance. This is especially true because the performance hinges greatly on the quality of the shift model since we only run one iteration of the DMD update.

## 3.3 Parameterizing with Normalizing Flows

In order to optimize the upper-level objective in Equation (8) with respect to $\lambda$, we need to be able to compute the density $\pi_{\theta, \lambda}$ directly. Therefore, we choose to represent $h_{\lambda}$ with a normalizing flow (NF) [16, 18, 17], which explicitly learns the density by defining an invertible transformation that maps latent variables to observed data. Generally, we compose a series of component flows together, i.e. $h_{\lambda} = h_{\lambda_K} \circ h_{\lambda_{K-1}} \circ \cdots \circ h_{\lambda_1}$, which define a series of intermediate variables $\hat{\boldsymbol{y}}_0, \ldots, \hat{\boldsymbol{y}}_{K-1}, \hat{\boldsymbol{y}}_K$, with $\hat{\boldsymbol{y}}_0 = \hat{\boldsymbol{z}}$ and $\hat{\boldsymbol{y}}_K = \hat{\boldsymbol{u}}$. The log-likelihood of the composed flow is given by:

$$\log \pi_{\theta, \lambda}(\hat{\boldsymbol{u}}|c) = \log p_{\theta}(\hat{\boldsymbol{z}}) - \sum_{i=1}^{K} \log \left| \det \frac{\partial \hat{\boldsymbol{y}}_i}{\partial \hat{\boldsymbol{y}}_{i-1}} \right|. \tag{9}$$

In this work, we make use of the affine coupling layer proposed by Dinh et al. [18] as part of the real non-volume-preserving (RealNVP) flow. The core idea is to split the input $\hat{\boldsymbol{u}}$ into two partitions $\hat{\boldsymbol{u}} = (\hat{\boldsymbol{u}}_{I_1}, \hat{\boldsymbol{u}}_{I_2})$, where $I_1$ and $I_2$ are a partition of $[1, MH]$, and apply

$$\hat{\boldsymbol{y}}_{I_1} = \hat{\boldsymbol{u}}_{I_1}, \quad \hat{\boldsymbol{y}}_{I_2} = \hat{\boldsymbol{u}}_{I_2} \odot \exp s_{\lambda}(\hat{\boldsymbol{u}}_{I_1}, c) + t_{\lambda}(\hat{\boldsymbol{u}}_{I_1}, c), \tag{10}$$

where $s_{\lambda}$ and $t_{\lambda}$ are the scale and translation terms, which are represented with arbitrary neural networks, and $\odot$ is the Hadamard product. This makes computing the log-determinant term in Equation (9) and inverting the flow fast and efficient. Now, in robotics, we often have lower and upper limits on the controls. These are usually enforced in sampling-based MPC by either clamping the control samples or passing them through a scaled sigmoid. However, instead of enforcing the constraints heuristically after sampling, we learn a constrained sampling distribution directly. Since the sigmoid function is invertible and has a tractable log-determinant (shown in Appendix A.8), we can simply append one after $h_{\lambda_K}$ in the NF and scale it by the control limits. This ensures that control constraints are satisfied by design and taken into account while learning the distribution.

## 3.4 Training the Sampling Distribution

Computing gradients through the upper-level objective is not straightforward, as both the expectation and the inner terms of Equation (7) depend on $\lambda$. Therefore, the state distribution depends on the NF and latent shift model. One way around this issue is to consider a modified objective at each batch $d$:

$$\ell_d(\boldsymbol{\theta}, \lambda; c) = \mathbb{E}_{\boldsymbol{\pi}_{\boldsymbol{\theta}, \lambda_d}, \rho, f} \left[ \sum_{t=0}^{T-1} \hat{J}(\theta_t, \lambda; x_t, c) \right], \tag{11}$$

which fixes the outer expectation to be with respect to the current policy. Intuitively, this choice trains the NF to optimize the MPC cost function under the state distribution resulting from the current policy $\boldsymbol{\pi}_{\boldsymbol{\theta},\lambda_d}$. We then update the outer expectation distribution at each episode.

Now, we only have to focus on computing the gradient $\nabla_\lambda \hat{J}(\theta_t(\lambda), \lambda; x_t, c)|_{\lambda=\lambda_d}$ for each time step, which can be computed similar to Equation (3) and approximated with Monte Carlo sampling:

$$\nabla \hat{J}(\theta_t(\lambda), \lambda; x_t, c) \approx - \sum_{i=1}^{N} w_i \nabla_\lambda \log \pi_{\theta_t(\lambda),\lambda}(\hat{\boldsymbol{u}}_t^{(i)}|c), \tag{12}$$

where the weights $w_i$ are defined according to Equation (5). The log-likelihood is given by Equation (9), the gradient of which involves computing the backwards pass through the network $h_\lambda$. However, we also have to consider the dependence of the latent distribution parameters $\boldsymbol{\theta}(\lambda)$ on $\lambda$. Therefore, we must backpropagate through the MPC update:

$$\boldsymbol{\mu}_t(\lambda) = (1-\gamma_t)\tilde{\boldsymbol{\mu}}_t(\lambda) + \gamma_t \Delta\boldsymbol{\mu}_t, \quad \Delta\boldsymbol{\mu}_t = \frac{\mathbb{E}_{\pi_{\tilde{\theta}_t(\lambda),\lambda}, \hat{f}}\left[e^{-\frac{1}{\beta}C(\hat{\boldsymbol{x}}_t, \hat{\boldsymbol{u}}_t)}h_\lambda(\hat{\boldsymbol{u}}_t; c)\right]}{\mathbb{E}_{\pi_{\tilde{\theta}_t(\lambda),\lambda}, \hat{f}}\left[e^{-\frac{1}{\beta}C(\hat{\boldsymbol{x}}_t, \hat{\boldsymbol{u}}_t)}\right]} \tag{13}$$

where the previous shifted mean $\tilde{\boldsymbol{\mu}}_t(\lambda)$ is given by the learned latent shift model. Note that we have rewritten the expectations in terms of the control distribution, rather than the latent distribution. This is necessary in order to derive the following approximate gradient without requiring differentiability. To compute the gradient of Equation (13), we must approximate the gradient of $\Delta\boldsymbol{\mu}_t$ with respect to $\lambda$, which we can compute as $\frac{\partial \Delta\boldsymbol{\mu}_t}{\partial \lambda} \approx M_1 - M_2 M_3$, where we define:

$$M_1 = \sum_{i=1}^{N} w_i \left[ \nabla_\lambda h_\lambda(\hat{\boldsymbol{u}}_t^{(i)}; c) + h_\lambda(\hat{\boldsymbol{u}}_t^{(i)}; c)\nabla_\lambda \log \pi_{\tilde{\theta}(\lambda),\lambda}(\hat{\boldsymbol{u}}_t^{(i)}|c) \right],$$

$$M_2 = \sum_{i=1}^{N} w_i h_\lambda(\hat{\boldsymbol{u}}_t^{(i)}; c), \quad M_3 = \sum_{i=1}^{N} w_i \nabla_\lambda \log \pi_{\tilde{\theta}(\lambda),\lambda}(\hat{\boldsymbol{u}}_t^{(i)}|c). \tag{14}$$

The derivation of this approximate gradient can be found in Appendix A.7. Note that computing the gradients $\nabla_\lambda \log \pi_{\tilde{\theta}(\lambda),\lambda}(\hat{\boldsymbol{u}}_t^{(i)}|c)$ will also require us to backpropagate through the shift model due to the dependence of $\tilde{\theta}$ on $\lambda$. Therefore, even when the step size is set to one, i.e. $\gamma_t = 1$, we introduce a form of recurrence between time steps. Please see Appendix A.1 for additional visualizations of the computational graph generated by an episode and further descriptions of the overall algorithm.

## 4 Related Work

Multiple works have considered sampling distributions beyond simple Gaussians, such as Gaussian mixture models [10] and a particle method based on Stein variational gradient descent (SVGD) [11]. Additionally, most implementations use heuristics to modify samples and squeeze out additional performance gains [4, 21]. In terms of learning the distribution, Amos and Yarats [12] learn a latent action space for their proposed differentiable cross-entropy method (CEM) controller. However, they require all components of the pipeline to be differentiable and do not consider learning a shift model. Agarwal et al. [13] learn a normalizing flow in the latent space of a variational autoencoder (VAE). Yet, they also require differentiability, use expert demonstrations to learn the latent space of the VAE, and have no means of warm-starting between time steps. Wang and Ba [22] propose to use a learned feedback policy to warm-start MPC, but still rely on a Gaussian perturbations to the proposed action sequence. The authors also explore performing online planning in the space of the network's parameters, which results in a massive action space that may be hard to scale.

Power and Berenson [14, 15] also train a normalizing flow to use as the sampling distribution for MPC, but they do not learn a latent shift model and perform all operations in the control space. Moreover, they mix the latent samples with Gaussian perturbations to the current control-space mean, as they do not update the latent distribution directly. This prevents them from fully taking advantage of the learned distribution and throws away useful information which could potentially improve performance. In fact, we show in Appendix A.4 that the learned shift model contributes significantly to the performance gains. Additionally, a primary focus of their work is on how to

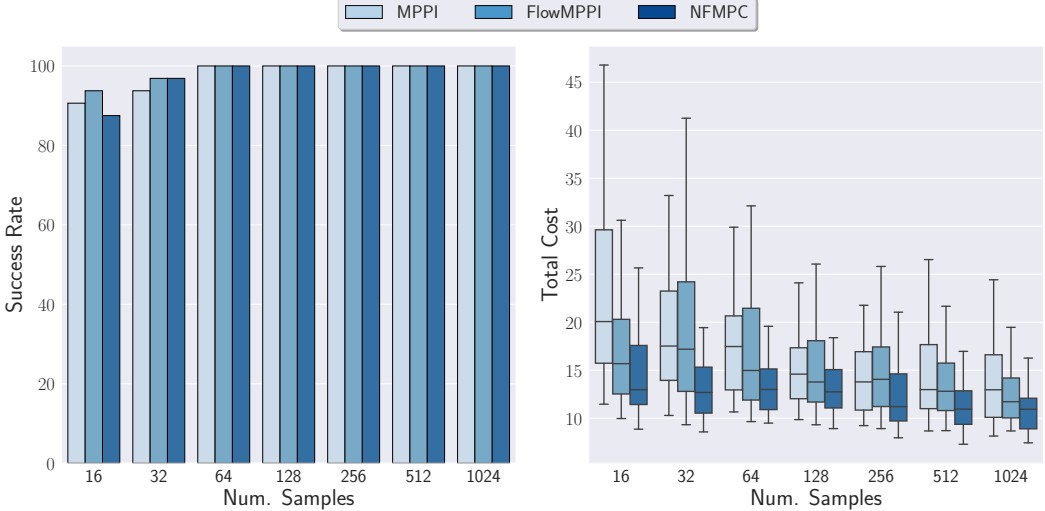

Figure 1: Success rate and cost distribution on the PNRANDDYN environment across a different number of samples.

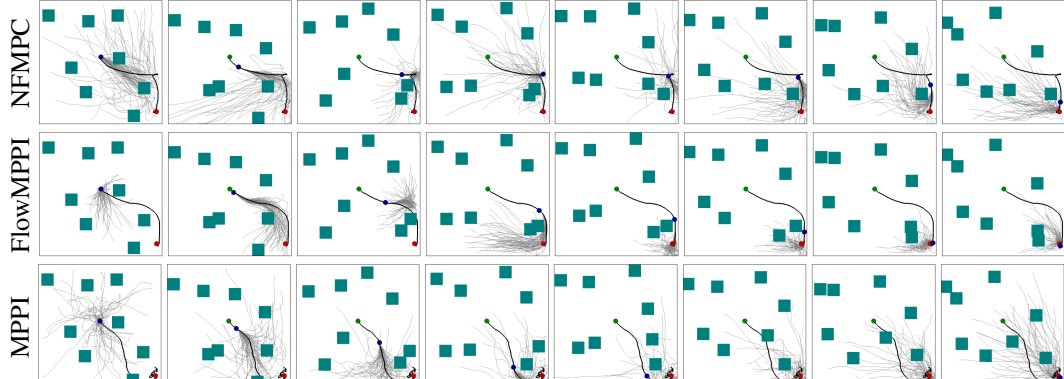

Figure 2: Visualization of a trajectory and top samples from (top) **NFMPC**, (middle) **FlowMPPI**, and (bottom) **MPPI** on the PNRANDDYN task.

handle out-of-distribution (OOD) environments by learning a posterior over environment context variables. We could combine their approach with ours by conditioning the learned shift model on the inferred environment context for improved generalization. Finally, normalizing flows have also been used to improve exploration in RL [23, 24, 25] by providing a more flexible, and potentially multimodal, distribution. They have also been employed in sampling-based motion planning [26, 27] to provide good proposal configurations to speed up convergence.

## 5 Experimental Results

In all experiments, we denote our proposed approach as **NFMPC**, the baseline MPPI implementation as **MPPI**, and the method by Power and Berenson [14, 15] as **FlowMPPI**. Details about the hyper-parameters, implementation, tasks, and training can be found in Appendix A.2. We evaluate on a fixed set of environments, which includes start states, goal locations, and obstacle placements, and run 32 rollouts for each sample amount. Our primary metrics for comparison are the success rate, defined as the percentage of times the task goal was achieved, and the average cost of trajectories which successfully completed the task. Additionally, we cannot ignore the overhead introduced by the NF in the control pipeline. Therefore, we measured the average change in wall clock time across different amounts of samples for **NFMPC** and **FlowMPPI** in Appendix A.5.

### 5.1 Planar Robot Navigation

We begin by applying **NFMPC** to a planar robot navigation task in which a 2D holonomic point-robot must reach a goal position while avoiding eight dynamic obstacles, which move around randomly at

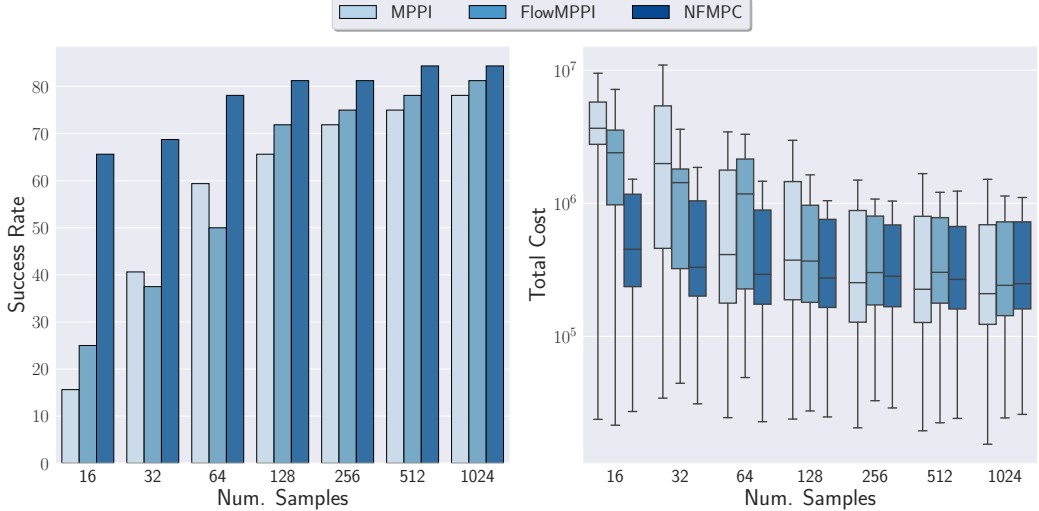

Figure 3: Success rate and cost distribution on the FRANKAOBSTACLES environment across a different number of samples.

each time step (PNRANDDYN). The point-robot has double integrator dynamics with stochasticity on the acceleration commands to create a mismatch between the predictive model used by MPC and the true environment. Each obstacle's current position is perturbed with Gaussian noise and clipped to be within map bounds, and the starting and goal locations of the robot are randomized in each episode. The NF for both **NFMPC** and **FlowMPPI** is conditioned on the obstacle locations, current state, and goal position. Note that we do not condition the shift model, as this consistently hurt performance.

We quantitatively compare all controllers in Figure 1. The trajectory cost box plots represent the median and quartiles of the distribution. We find that **NFMPC** consistently outperforms the both **MPPI** and **FlowMPPI**. While all controllers reach a 100% success rate at 1024 samples, **NFMPC** achieves a 20% and 8% lower median cost over **MPPI** and **FlowMPPI**, respectively. We also find that **NFMPC** scales more gracefully overall as the number of samples is reduced. For instance, it is able to withstand a $32\times$ decrease in the number of samples (1024 to 32) while reducing success rate by only 3% and increasing median cost by 8%, nearly matching **FlowMPPI** at 1024 samples. In comparison, **MPPI** reduces success rate by 6% and increases median cost by 43%. Similarly, **NFMPC** outperforms **FlowMPPI** at all sample amounts in terms of median cost, although at 16 samples **FlowMPPI** achieves a slightly higher success rate than **NFMPC**. We visualize trajectories and top samples in Figure 2. The green and red dots are the starting and goal locations, respectively, and the blue dot is the current position of the robot at the given time step. The thick black line is the resulting path taken by the controller, while the gray lines are the top samples generated at the current state. In this example, **FlowMPPI** nearly collides with an obstacle, while **NFMPC** and **MPPI** are able to safely reach the goal. **FlowMPPI** over commits to a narrow corridor and is unable to reroute in time to account for the new obstacle location. **NFMPC** takes a similar trajectory to **FlowMPPI**, however, it is able to pause until the obstacle moves out of the way to proceed towards the goal. We also provide additional experiments with static environments and comparing unconditional and conditional models in Appendix A.3.

### 5.2 Franka Panda Arm

Next, we apply **NFMPC** to the FRANKA task, which involves controlling a 7 degree-of-freedom (DOF) Franka Panda robot arm and steering it towards a randomly placed target goal from a fixed starting pose while avoiding a single pole obstacle. The NF for both **NFMPC** and **FlowMPPI** is conditioned on the obstacle locations, initial state, and goal position. It is also important to note that no controller achieves a 100% success rate, as not every randomly generated environment is feasible. As shown in Figure 3, at 1024 samples, **NFMPC** achieves a success rate of 84%, while **FlowMPPI** and **MPPI** only succeed 81% and 78% of the time, respectively. Again, **NFMPC** scales better with a reduced number of samples. With a $64\times$ decrease in the number of samples (1024 to 16), **NFMPC** only drops in success rate by 29%. Meanwhile, **FlowMPPI** decreases by nearly 70% to a success rate of 25%. However, both fair better than **MPPI**, which drops by 80% to a success rate of 16%.

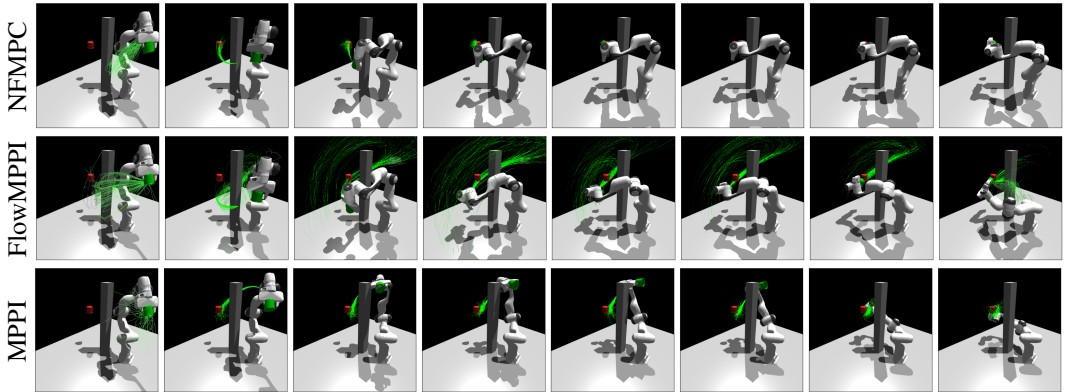

Figure 4: Visualization of a trajectory and top samples from (top) **NFMPC**, (middle) **FlowMPPI**, and (bottom) **MPPI** on the FRANKAOBSTACLES task.

These results support the hypothesis that learning to perform MPC updates in the latent space of the NF and training the controller as a recurrent network improves performance.

To more clearly understand what **NFMPC** is doing differently, we visualize the performance of all three controllers on a held-out validation environment in Figure 4. The green and red markers indicate the end-effector and goal positions, respectively, while the single pole is the obstacle which must be avoided. Additionally, the green trajectories represent the top samples from the controller at each time step. While **MPPI** collides with the obstacle, both **NFMPC** and **FlowMPPI** learn to take a different path which is collision-free. **FlowMPPI** generates better initial trajectories than **NFMPC**, which are more pointed towards the goal location and achieve a greater velocity. However, **NFMPC** is able to better adapt the sampling distribution throughout the episode and reach the goal more quickly. Finally, in Appendix A.4, we present results for the environment with no obstacles, a comparison of conditional and unconditional models, and a breakdown of the individual cost terms for all models to gain insight into the learned sampling distributions. We also perform an ablation which removes the learned shift model from **NFMPC** to illustrate that it is a crucial component.

## 6 Limitations

A major limitation of **NFMPC** is that the learned distribution and shift model are only valid for a fixed horizon and control dimensionality. Therefore, these components cannot be directly transferred to new robots or for alternate horizons without being retrained. However, this could potentially be remedied by novel architectural innovations and training distributions across both environments and robots. Moreover, the learned distribution is specific to the environmental distributions on which it was trained. Therefore, it does not always perform as well when transferred to out-of-distribution environments. However, this is always going to be a challenge for any learning-based method and addressing it is an open question for future research. Finally, both our approach and **FlowMPPI** introduce an additional overhead to due to running the NF which cannot be ignored. This increase in wall clock time may be worth the additional performance gains. Additionally, we scale better with a reduction in samples compared to all baselines. Therefore, we can potentially alleviate some of the introduced overhead by reducing the number of samples while still meeting the application demands.

## 7 Conclusion

We presented a method for learning MPC sampling distributions with normalizing flows (NFs) which moves all online parameter updates and warm-starting operations into the latent space. We show how to frame the problem as bi-level optimization and derive an approximate gradient through the MPC update to train the distributions. Additionally, we illustrate how to incorporate control box constraints directly into the NF architecture. Through our empirical evaluations in both simulated navigation and manipulation problems, we demonstrate that our approach is able to surpass the performance of all baselines. Moreover, we find that controllers which move all operations into the latent space are often able to scale more gracefully with a reduction in the number of samples. These results indicate the importance of leveraging the latent space in learned sampling distributions for MPC. Finally, because we learn all components of the controller through episodic interactions with the environment, they can potentially be trained to account for the modeling errors in the MPC controller.

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
