# OpenReview forum: "Learning Sampling Distributions for Model Predictive Control"
_robot-learning.org/CoRL/2022/Conference — CoRL 2022 Poster_

### Official Review · Reviewer_bXam · 2022-07-24

**Originality:** Good
**Technical Quality:** Good
**Clarity Of Presentation:** Very Good
**Impact:** 3

**Recommendation:**

Weak Accept: I recommend accepting the paper, but will not argue for my recommendation if the majority of other reviewers have a different opinion.

**Summary:**

This paper presents an approach to learning sampling distributions for MPPI. The authors leverage normalizing flows, and backpropagate through the MPC optimization procedure to train the flow model.

**Issues:**

As discussed above, the main issue to address is OoD tasks, as well as several more minor issues such as changing the framing around the generative model.

**Quality Of The Limitations Section:**

Limitations are addressed clearly

**Reviewer Expertise:**

4: The reviewer is confident but not absolutely certain that the evaluation is correct

**Robotics Focus:**

Highly relevant to robotics but no hardware experiments

**Strengths And Weaknesses:**

## Strengths

Overall, I think the paper is a good one. The method is straightforward, easy to understand, and clearly yields strong practical benefits.

## Weaknesses

There are a handful of small issues which, if addressed, would substantially strengthen the paper.

- The biggest issue is lack of investigation of OoD tasks. The authors explicitly discuss this in their limitations section, but it really is extremely important. The paper needs to investigate performance on OoD tasks and should include more complex environments (possibly featuring multiple obstacles). In general, the authors should investigate leveraging more information in the normalizing flow. For example, scaling to alternative, more complex environments would likely be substantially improved by conditioning the NF on environmental information (which as far as I can tell is not currently done). This approach has been widely investigated in the literature on learning sampling distributions for sampling-based motion planning, and I would point the authors toward that area for inspiration (the authors should probably also discuss that literature in their related work section, as it is highly related).
- The authors should report success/cost versus wall clock time in addition to versus number of samples. The performance for a small number of samples is obviously much better than standard MPPI, but understanding the time cost of the sampling (especially for NF models) is important.
- It is unclear if the action sequence samples are decoded/passed through the NF as iid samples or jointly. If each set of action sequences is decoded independently, investigating a joint distribution over all samples is potentially very interesting as the model would be able to learn variance reduction strategies (eg antithetic sampling).
- The authors propose to use NF models primarily due to the fact that they enable exact evaluation of likelihoods. Generally, with the world of generative models evolving as quickly as it currently is, I would try to minimize the explicit dependence on the NF models. In particular, the authors currently state a set of necessary features for the learning model in several spots across the paper. The paper would be improved by formally stating, in one spot, the necessary features for a generative models to be used in this work, and then discuss how NF models are one example of a model that meets the desiderata whereas eg VAEs are not. This hopefully will improve the applicability of this paper as generative models continue to progress.

**Summary Of Recommendation:**

The authors should extend their method to OoD tasks, as well as improve some elements of the writing. I am willing to improve my impact score from 3 to 4 if there is (substantial) progress made in the OoD task area.

---

> ### Author Response · Authors · 2022-08-23
> **Response to Reviewer bXam**
>
> We thank the reviewer for their feedback and pointing out a number of issues and questions about the paper which need further clarification. We address each issue below and will include these discussions in the final paper.
>
> ### Out-of-Distribution Tasks
> Testing on out-of-distribution (OoD) tasks is always going to be a challenge for any learning-based method. Part of the advantage to data-driven approaches lies in their ability to leverage the structure in the training data. In this paper, our focus was to demonstrate that there is substantial structure in robotics tasks for a learning-based method to exploit and gain significant performance benefits for sufficiently similar environments. However, we agree that finding ways of adapting to OoD environments is a relevant and important direction of research which we will explore in future work.
>
> ### Conditioning on Environmental Information
> Indeed, we agree that conditioning on environment information is important and would help scale the method to more complex environments. In fact, we have actually done some initial experiments exploring this in the setting of Planar Robot Navigation in Appendix A.3. Specifically, we define two new tasks: 1) PNRand, in which eight obstacles are randomly placed in the environment instead of a fixed grid, and 2) PNRandDyn, a dynamic version of PNRand in which the obstacles move around randomly each time step. In both cases, we consider conditioning the normalizing flow (NF) on obstacle locations, initial state, and goal position. Unlike in our main experiments, we found that conditioning the NF actually improved performance over the unconditional model. This proved to be especially useful in the case of dynamic obstacles. However, we agree with the reviewer that this warrants even further work than these preliminary results. We aim to explore conditioning on environmental information on more complex environments and tasks in future work. Specifically, we are currently exploring setting up NFMPC on a real Franka arm operating in an environment with multiple obstacles and in an off-road autonomous driving task.
>
> ### Impact of the Normalizing Flow on Timing
> We agree that measuring the impact on wall clock time is an important metric missing from the current paper. The introduction of the NF does have a nontrivial impact on timing. However, NFMPC has a number of runtime benefits compared to FlowMPPI and MPPI run for more than a single iteration per time step. Please refer to our response to Reviewer BrmL for a more detailed breakdown and discussion.
>
> ### Non-IID Sampling
> Currently, all latent samples that are passed through the NF are sampled independently from the latent Gaussian. However, we agree that exploring non-IID sampling methods is an exciting future direction to build on this work. It may allow us to further improve performance with fewer samples.
>
> ### Necessary Features of the Learned Distribution
> We completely agree that the paper would be strengthened by a more concise and formal discussion of the necessary features for a generative model to be used as the learned sampling distribution. The two main criteria are that the generative model has a tractable likelihood and that it is feasible to update at least a subset of its parameters online. This is why, for instance, we do not use VAEs, as they do not allow for an exact evaluation of likelihood, only a lower bound. We will use this suggestion to improve the overall final manuscript and elaborate on this point.

---

### Official Review · Reviewer_aRE3 · 2022-07-24

**Originality:** Good
**Technical Quality:** Very Good
**Clarity Of Presentation:** Good
**Impact:** 3

**Recommendation:**

Weak Accept: I recommend accepting the paper, but will not argue for my recommendation if the majority of other reviewers have a different opinion.

**Summary:**

This paper proposes improves previous works on model-predictive control based on normalizing flow. The previous works learn the sampling distribution in the original control space. In contrast, this paper proposes to perform the learning of the sampling distribution in the latent space. As a result, the entire learning algorithm can be cast as a bi-level optimization problem which alternately optimize the sampling distribution and the normalizing flow which is responsible for mapping a sample in the latent space to the control space.

**Issues:**

Please see the question under "Weaknesses" above.

**Quality Of The Limitations Section:**

Limitations are addressed clearly

**Reviewer Expertise:**

2: The reviewer is willing to defend the evaluation, but it is quite likely that the reviewer did not understand central parts of the paper

**Robotics Focus:**

Highly relevant to robotics but no hardware experiments

**Strengths And Weaknesses:**

Strengths:
- The proposed improvement of updating the latent sampling distribution is reasonable and solved in an intuitive manner.
- The paper is in general well written, although I have trouble understanding some of the technical details such as Section 3.4, but it may be due to my lack of knowledge in the relevant fields such as normalizing flow.

Weaknesses:
- The experiments indeed show the improvements of the proposed method over the two baselines under comparison, however, I feel that some of the experimental results need more explanations. For example, in all experiments, it seems that the method of FlowMPPI usually performs worse than the simple baseline of MPPI especially with small sample sizes (e.g., see lines 246-248). Why is this the case? Next, as said in lines 251-252, the proposed NFMPC is able to discover more diverse paths, so what's the intuition behind this improved behavior of NFMPC? Is it because updating the sampling distribution in the latent space, as done by the proposed method, helps learn a better sampling distribution?

**Summary Of Recommendation:**

This paper makes a reasonable and useful contribution of sampling-based model predictive control.

---

> ### Author Response · Authors · 2022-08-23
> **Response to Reviewer aRE3**
>
> We thank the reviewer for their feedback and highlighting areas in the evaluation which need further explanation and analysis. We elaborate on all of your concerns below and will include these discussions in the final paper.
>
> ### Performance of FlowMPPI with Small Sample Sizes
> We did find that FlowMPPI often performs worse than the MPPI baseline with small sample amounts. In the standard implementation of FlowMPPI, half of the samples come from the normalizing flow (NF) and the other half are Gaussian perturbations of the current control-space mean. Initially, the samples coming from the NF provide a good initialization for the control distribution mean. However, as the latent Gaussian distribution used by the NF is never updated, half of our samples are always coming from this same distribution. Therefore, our hypothesis is that as the robot moves in the environment, these samples may cease to be as useful or informative. We then have to rely on the other half of samples coming from Gaussian perturbations of the control-space mean to do most of the work. As such, we effectively have half the budget of samples to work with than MPPI would, as the samples from the NF potentially do not provide much useful information. At higher sample amounts, there are still enough samples available for FlowMPPI to adapt the mean effectively, and its improved performance over MPPI is potentially due to its good initialization. However, at lower sample amounts, there are not enough samples available for FlowMPPI to effectively adapt to disturbances in the environment, mismatch between the model and the true environment, and potentially myopic decisions due to a shorter horizon.
>
> ### Improved Behavior of NFMPC
> We agree with your intuition as to why NFMPC performs better. It is our hypothesis that performing all operations in the latent space of the NF allows us to take better advantage of the learned sampling distribution. Moreover, with our approximate gradient discussed in Section 3.4, we are able to train the NF by backpropagating through the MPPI update of the latent mean. This allows us to optimize the controller directly for performance on the desired task. Additionally, it allows us to shape the latent space such that performing the MPPI update of the latent mean achieves good performance. Furthermore, we jointly learn the NF along with the latent shift model. As the usual method of warm-starting, which simply shifts the current plan forward in time, may be sub-optimal, learning the shift model is another source of potential performance improvement. In fact, we perform an ablation which removes the learned shift model in Appendix A.4. We find that removing the learned shift model from a pre-trained NF hurts performance. Additionally, we find that training a NF without a shift model results in even worse performance than simply removing it from the pre-trained controller. This indicates that the inclusion of a learned shift model and training the entire controller with backpropagation-through-time potentially allows it to discover lower-cost trajectories to the goal.

---

### Official Review · Reviewer_BrmL · 2022-08-01

**Originality:** Good
**Technical Quality:** Good
**Clarity Of Presentation:** Good
**Impact:** 3

**Recommendation:**

Weak Accept: I recommend accepting the paper, but will not argue for my recommendation if the majority of other reviewers have a different opinion.

**Summary:**

This work uses bi-level optimization and a normalizing flow parameterization of learned sampling distributions. This allows for learned sampling distributions for MPC without requiring differentiability of dynamics and cost functions. This approach allows for box constraints on the controls (ex joint limits).

The proposed method is compared to MPPI, FlowMPPI on a 2d navigation task, and a 7dof reaching task with a Frank robot.



**Issues:**


My main concern is that number of samples is an insufficient metric to compare this method to existing baselines. If your approach only requires 16 samples, while MPPI requires 1024, how much faster is your method or how much less computation is required?

**Quality Of The Limitations Section:**

Limitations are addressed clearly

**Reviewer Expertise:**

3: The reviewer is fairly confident that the evaluation is correct

**Robotics Focus:**

Highly relevant to robotics but no hardware experiments

**Strengths And Weaknesses:**

Strengths:
The proposed approach appears to scale more gracefully to smaller numbers of samples, particularly for the harder Franka & Franka Obstacles tasks.

Weakness:
I still don't have a good intuition for the average trajectory cost in terms of either wall clock time or required number of operations for inference. This work demonstrates that their approach can operate with significantly fewer samples than MPPI, but is unclear what the advantage of less samples actually implies. How much faster or slower is it to run this NF based approach vs MPPI? Does the 16-sample NFMPC run any faster or require less compute than a 1024 sample MPPI?

There maybe clear benefits to the proposed approach here, that this manuscript would benefit from if articulated. Alternatively, if the proposed approach is actually slower or more computationally expensive even when using less samples, this work would be less compelling.

**Summary Of Recommendation:**

The paper is well written, figures are clean and the ideas are well presented.  It would be great if authors could provide some insight into the issues section to help understand the impact of this work.

---

> ### Author Response · Authors · 2022-08-23
> **Response to Reviewer BrmL**
>
> We thank the reviewer for their feedback and pointing out an important comparison point that was missing from the main paper, which we address below.
>
> ### Impact of the Normalizing Flow on Timing
> We measured the average change in wall clock time across different amounts of samples for NFMPC and FlowMPPI compared to the baseline MPPI implementation on our NVIDIA Titan V GPU. For the Planar Robot Navigation and Franka Panda Arm experiments, the average change is 1.61X and 1.91X, respectively. Moreover, compared to MPPI with 1024 samples, the change in wall clock time for NFMPC and FlowMPPI with 16 samples is approximately 1.01X and 1.14X, respectively. Therefore, the introduction of the normalizing flow (NF) has a notable impact on wall clock time for both methods. However, this overhead is expected and does not prohibit either method’s utility in the real world. And there are still clear performance advantages of NFMPC over the baselines in terms of success rate and average trajectory cost for a given sample amount.
>
> Furthermore, the performance of MPPI reported in the paper and the above timing comparisons are when the optimization is run for a single iteration per time step. However, the performance of MPPI generally improves with an increased number of iterations at the cost of an increased runtime. For instance, we compare the performance of MPPI run for 3 iterations per time step with NFMPC run for a single iteration, both using 1024 samples. In the Franka task (no obstacles), we can reduce the average trajectory cost of MPPI to be only 12% worse than NFMPC, rather than the previous 19%. When we introduce obstacles for the FrankaObstacles task, MPPI with 3 iterations can actually match the success rate of NFMPC, which is 87.50%, albeit with a worse average trajectory cost. However, in this case, NFMPC actually results in an average **reduction** of wall clock time by 24.8%. Similarly, for the PNGrid task, MPPI with 3 iterations reduces the average trajectory cost to be only 10% worse than NFMPC, rather than the previous 40%. However, this again comes at the cost of increased runtime, as NFMPC reduces the average wall clock time by 26.8%. Therefore, NFMPC allows us to surpass the performance of MPPI run with more iterations while reducing the required runtime.
>
> Additionally, NFMPC has substantial runtime benefits over FlowMPPI due to its improved scaling. For the PNGrid task, NFMPC with 128 samples outperforms FlowMPPI with 1024 samples while reducing average runtime by 22%. Similarly, for the Franka task, NFMPC with 64 samples outperforms FlowMPPI with 1024 samples while reducing average runtime by 43%. And for both Franka tasks, NFMPC with 16 samples outperforms FlowMPPI with 128 samples while reducing average runtime by 8%. As such, there are clear runtime benefits for NFMPC over both FlowMPPI and MPPI run for additional iterations. We will include all of these additional timing results and comparisons in the final paper. Finally, it is also important to note that we did not perform a hyperparameter sweep on the normalizing flow, and it may be possible to significantly reduce the size of the network while retaining performance benefits.

---

### Meta-Review · Area_Chair_DhsY · 2022-08-14

**Recommendation:** Accept (Poster)
**Confidence:** 4

**Metareview:**

Phase 1:

Strengths:
The submission is overall seen as providing intuitive technical contributions and and clear presentation. The experiments support the introduction of NFs into MPC and the method works well will reduced number of samples.

Weaknesses:
The evaluation is missing out on some important details such as a comparison with respect to compute or wall time in addition to pure sampling numbers (NFs introduce additional computational cost which cannot be ignored). Some experimental results could be analyzed in further detail (e.g. low performance of FlowMPPI). Importantly, additional experiments are requested to evaluate how well the learned distribution generalizes to OOD tasks.

Phase 2:

The feedback has been generally positive (3 weak accepts) with only minor criticism. In particular, the reviews appreciate intuitive technical contributions and clarity of presentation but criticize some missing aspects of the evaluation. I agree with the reviewers and recommend acceptance. Please take the remaining points from the review process seriously and follow up with improvements on open points and promised changes.

**Best Paper Nomination:**

No